# JaxPruner: A Concise Library for Sparsity Research

Joo Hyung Lee, Wonpyo Park, Nicole Mitchell, Jonathan Pilault, Johan Obando-Ceron,
Han-Byul Kim, Namhoon Lee, Elias Frantar, Yun Long, Amir Yazdanbakhsh, Shivani Agrawal,
Suvinay Subramanian, Xin Wang, Sheng-Chun Kao, Xingyao Zhang, Trevor Gale, Aart Bik,
Woohyun Han, Milen Ferev, Zhonglin Han, Hong-Seok Kim, Yann Dauphin
Gintare Karolina Dziugaite, Pablo Samuel Castro, Utku Evci

**Google Research**

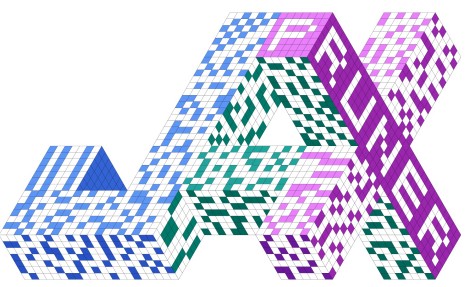

This work introduces *JaxPruner*, a JAX-based sparsity library for machine learning research. *JaxPruner* aims to accelerate research on sparse neural networks by providing concise implementations of popular pruning and sparse training algorithms with minimal memory and latency overhead. Algorithms implemented in *JaxPruner* share a common API and works seamlessly with Optax, a widely-used optimization library in JAX, which further enables easy integration with other JAX-based libraries. We demonstrate the ease of integration by providing examples in four different codebases: Scenic, t5x, Dopamine and FedJAX and provide baseline experiments on popular benchmarks. *JaxPruner* is hosted at github.com/google-research/jaxpruner

## 1. Why a new sparsity library in JAX?

Sparsely connected neural networks have shown to achieve better performance than dense models with the same parameter count [1, 2]. However, utilizing sparsity and realizing its potential in realistic scenarios requires a close collaboration between hardware, software and algorithms research. To this end, it often requires a flexible code library to enable rapid prototyping of ideas and evaluating them on a variety of ever-changing benchmarks.

Over the last few years, JAX [3] has seen increasing adoption by the research community [4–7]. The key difference between JAX and other popular deep learning frameworks such as PyTorch [8] and Tensorflow [9] is the clear separation between functions (e.g. neural networks) and states (e.g. parameters). This makes function transformations like taking gradients, Hessian calculations or vectorization[1] relatively easy, thus reducing the time required for implementing complex ideas [10]. Similarly, having the entire state of a function isolated under a single dictionary makes it easy to modify and transform. As we will shortly discuss, these features also ease the implementation of common subroutines across different algorithms used in sparsity research.

Though implementations of individual algorithms with different sparsity structures exist [11, 12]), there is no comprehensive library for sparsity research in JAX. This motivated us to develop *Jax-Pruner*. There are two high-level strategies for achieving parameter sparsity: (1) **pruning** which aims to obtain sparse networks starting from dense networks for inference efficiency and (2) **sparse training** which aims to train sparse networks from scratch, thus reducing training cost as well. *Jax-Pruner* implements key baselines for each family of algorithms and makes it easy to extend them.

---

[1] https://jax.readthedocs.io/en/latest/jax-101/03-vectorization.html

First Conference on Parsimony and Learning (CPAL 2024).

In what follows, we discuss key design principles of *JaxPruner* (Section 2), provide a short overview of the library (Section 3) and share our results with baseline pruning and sparse training algorithms in (Section 4). We conclude with our plans for future versions.

## 2. Tenets: Fast Integration, Research First and Minimal Overhead

We want *JaxPruner* to facilitate sparsity research by providing strong baselines, making them easy to extend. Furthermore, we want algorithms in *JaxPruner* to work seamlessly in different libraries. We were guided by three tenets when designing the library in order to achieve these goals:

**Fast Integration**  Research in Machine Learning (ML) is fast paced. Combined with the huge variety of ML applications, this results in a high number of ever-changing codebases. At the same time, adoptability of new research ideas is highly correlated with their ease of use. For these reasons, we aimed to reduce friction for those integrating *JaxPruner* into an existing codebases by using the popular Optax optimization library [10]. State variables (i.e. masks, counters) needed for pruning and sparse training algorithms are stored together with the optimization state, which makes parallelization and checkpointing easy.

**Research First**  Often research projects require running multiple algorithms and baselines, and so they benefit greatly from rapid prototyping. *JaxPruner* achieves this by committing to a generic API shared among different algorithms, which in turn facilitates switching between algorithms. We provide well-documented implementations of common baselines, which facilitate modifications. Furthermore, we have made it easy to switch between common forms of sparsity (unstructured, N:M, block, etc.). A quick overview of such features is discussed in the next section.

**Minimal Overhead**  There are a growing number of options for accelerating sparsity in neural networks (e.g. N:M sparsity [13], CPU-acceleration [14], activation sparsity [15]). However, integration with existing ML frameworks is often lacking, making these advances relatively difficult to use, especially in research. Given our main goal of facilitating research, *JaxPruner* follows the tradition of using binary masks for representing sparsity, which introduces some additional operations and requires additional storage. We minimize this memory and run-time overhead by compressing mask variables. Furthermore we optimize the top-k functions used frequently in sparse training and pruning algorithms to reduce the run-time overhead. We also provide examples for using the experimental sparsity feature in JAX[2].

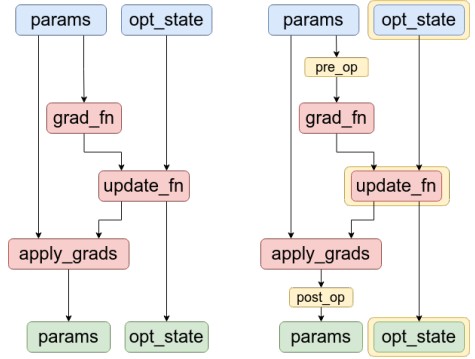

Figure 1: **Visualization of a training loop.** (left) common training loop (right) *JaxPruner* wraps the existing Optax transformations to store and update variables like masks needed for pruning. Additional operations (PRE/POST-OP) are added to modify parameters at different points.

## 3. Overview

*JaxPruner* consists of about 1000 lines of code (+850 lines of tests), organized into six modules. The library also includes interactive Python notebooks and integration with popular research libraries. Here we give a short overview of the *JaxPruner* API and list its key features.

**Optax Integration**  State-of-the-art pruning algorithms often require iterative adjustments to the sparsity mask used. Such iterative approaches are stateful, i.e., they require some additional variables like masks, counters, initial values, etc. This is similar to common optimization algorithms like Adam [16] and Momentum SGD, which require their optimization state to be handled throughout training. The majority of codebases in JAX achieve this through Optax, which bundles all variables of the optimization state as a parameter tree. A simplified diagram of a neural network training loop in JAX is given in Figure 1. At every step of the training, parameters and optimizer state are transformed using the gradients calculated through back-propagation. The Optax `update_fn` is used to

---

[2]https://jax.readthedocs.io/en/latest/jax.experimental.sparse.html

transform the gradients and the optimizer state. Finally, the resulting gradients are added to the parameters.

*JaxPruner* exploits the observation that most iterative pruning and sparse training algorithms can be thought of as special kinds of optimizers, which confine parameters into a sparse sub-domain. This key observation motivates us to use Optax gradient transformations to implement our algorithms. This approach reduces boiler-plate code required to integrate *JaxPruner* to existing codebases (e.g. checkpointing and handling mask variables). Below we give an example usage of *JaxPruner* inside an existing training loop and visualize these changes in Figure 1.

```
1  import jaxpruner
2
3  tx, params = _existing_code()
4  pruner = jaxpruner.MagnitudePruning(...) # Line 1: Create pruner.
5  tx = pruner.wrap_optax(tx) # Line 2: Wrap optimizer.
6
7  opt_state = tx.init(param)
8  # Line 3: [Optional] modifies weights temporarily for the forward pass.
9  forward_params = pruner.post_gradient_update(forward_params, opt_state)
10 new_params, new_opt_state = _training_step(tx, opt_state, forward_params)
11 # Line 4: Apply masks to parameters.
12 new_params = pruner.post_gradient_update(new_params, new_opt_state)
```

**One-shot Pruning** Most iterative pruning algorithms can be converted into one-shot pruning algorithms and vice-versa. Similarly, one can use pruning algorithms outside of a training loop. In order to address such this use case we include the `instant_sparsify` method in our API. `instant_sparsify` supports variable collections and individual JAX arrays. Below we give an example.

```
1  pruner = jaxpruner.MagnitudePruning(...) # Line 1: Create pruner.
2  X_pruned = pruner.instant_sparsify(X) # Line 2: Prune parameters.
```

**BaseUpdater** Most pruning or sparse training algorithms share the following routines: (1) initialize masks (2) apply masks and (3) update masks. This motivates us to unify common pruning and sparse training operations under a single stateless class: `jaxpruner.BaseUpdater`. BaseUpdater implements most of the API functions (like `wrap_optax`, and `instant_sparsify`) in a modular way such that different pruning algorithms can be implemented by overwriting only a few functions. This makes the extension of common pruning or sparse training algorithms relatively easy, reducing friction when trying new ideas. The BaseUpdater class is also highly customizable, in what follows we present three ways of controlling the behaviour of our algorithms.

**Custom Sparsity Distributions** In the pruning literature, it is common to apply custom sparsity levels to some of the layers, sometimes keeping them dense [17, 18]. We implement some of the most common sparsity distributions like *uniform* and *erk*. These distributions can be customized further by passing a mapping between individual parameters and their target sparsity values. Alternatively, users can define their own distribution functions and pass them to *JaxPruner* algorithms directly.

**Update Schedules** Most pruning algorithms work during training, but differ in how frequently they increase the sparsity, and when they apply the masks to parameters. Similarly, sparse training algorithms often require changes in the sparsity pattern at different frequencies. We implement common scheduling functions: one-shot, periodic, polynomial schedule [19]. Similar to sparsity distributions, users can define their custom schedules easily and pass them to the existing algorithms.

**Structured Sparsity** Despite exciting developments [14, 20], the challenge of accelerating *unstructured* sparse neural networks remains due to irregular memory access. Using more regular sparsity patterns like block [21–24] and N:M sparsity [13, 25–27] reduces irregular memory access and thus makes acceleration easier. However, networks with structured sparsity often perform worse compared to unstructured sparsity. Reducing this performance gap through better sparsity structures and algorithms is an active area of research. To this end, we make the sparsity type easy to customize by including common sparsity structures like Block and N:M sparsity [28]:

```
1  pruner = jaxpruner.MagnitudePruning(sparsity_type=jaxpruner.NbyM(2,4))
2  X_pruned = pruner.instant_sparsify(X)
```

**Other Features** We use `uint8` types for storing masks to reduce the memory footprint of our algorithms. For example, mask variables can increase the peak memory usage for training ViT-B/16 by about 3.9% ($6592 \rightarrow 6850$ MiB) when the batch size is 32. Since masks are binary variables (i.e. 0 or 1), they can be compressed further to reduce the memory footprint, which we support via the `use_packed_masks` flag. When packed masks are enabled the memory overhead reduces to 0.32% ($6592 \rightarrow 6613$ MiB) for the ViT setting just mentioned. We also provide an example that converts masked dense parameters of a pruned ViT-B/16 model to a sparse BCOO format and runs the model with significantly lower memory footprint using `jax.experimental.sparse`.

# 4. Baselines

Though sparsity research historically focused on computer vision benchmarks, there is a growing interest and need for using a more diverse set of domains when evaluating our research. To serve this goal, we provide integrations with some of the popular JAX libraries from different domains and benchmark algorithms implemented in *JaxPruner*. Spefically, we integrate *JaxPruner* with Scenic [29], T5x [30], Dopamine [31] and FedJAX [6]. Typically this requires changing only a few lines of code in the training loop as shown in the previous section.

Our unified API enables easy experimentation with a wide variety of algorithms. We implement the following set of algorithms as a representative set of baselines:

1. **Gradual Pruning** with random (RAND), saliency (SAL) [32] and weight magnitude (MAG) [19] criteria. We also implement global pruning with the weight magnitude criterion where pruning criterion is applied to all parameters at once (MAG-G). For global magnitude pruning we normalize the parameters of each layer before flattening using the L2 norm.

2. **Straight Through Estimator** with top-$k$ weight magnitude selection (STE). In sparse training with straight through gradients [33], parameters are projected into a sparse sub-space before the forward pass. Then gradients are calculated for all parameters and applied to the original set of dense parameters. STE is often applied using a fixed sparsity from the start of the training. In our experiments, however, we use the polynomial schedule used by the gradual pruning algorithms [19], as we observed this to give better results.

3. **Sparse Training** including static sparse training (STATIC) and dynamic sparse training with random (SET) [18] and gradient based (RIGL) [34] growth. In all of our experiments we use an initial drop fraction of 0.1 and apply cosine decay [35].

We benchmark pruning and sparse training algorithms in 4 different domains and discuss them in subsequent sections:

- (Section 4.1) ImageNet-2012 [36] image classification using the ViT-B/16 [37], PlainViT-S/16 [38] and ResNet-50 [39] architectures.
- (Section 4.2) Federated EMNIST [40] character recognition using a CNN with dropout [41].
- (Section 4.3) C4 language modelling using the T5-Base encoder-decoder transformer architecture [42, 43].
- (Section 4.4) a DQN agent [44] with a convolutional backbone trained on the MsPacman Atari 2600 game [45].

We share our results in Table 1. These baseline results provide a solid starting point for new research projects. Finally in Section 4.5, we re-visit some of the popular questions in sparsity research and run experiments using different sparsity distributions and structures.

## 4.1. Image Classification

We apply *JaxPruner* algorithms to train 80% sparse ViT-B/16, PlainViT-S/16 (PViT) and ResNet-50 models. Our goal in these experiments is not to get state-of art results. Instead, we aim to provide some baseline results using different training recipes and architectures to showcase the flexibility of *JaxPruner*. For all experiments, we use the default hyper-parameters provided by the Scenic library.

ResNet-50 is a popular architecture in sparsity literature [46, 47]. We train 80% sparse ResNet-50 models on ImageNet to reproduce previous results reported in the literature [17, 34]. We use

|        | T5-Base ($\downarrow$) | DQN | ResNet-50 | ViT-B/16 | ViT-B/16+ | PViT-S/16+ | Fed. MNIST |
|--------|---------|-----|-----------|----------|-----------|------------|------------|
| Dense  | 2.57 ±0.00 | 2589 ±503 | 76.60 ±0.12 | 73.94 ±0.11 | 74.71 ±0.27 | 80.11 ±0.07 | 86.21 ±0.39 |
| Rand   | 3.28 ±0.01 | 1435 ±381 | 70.31 ±0.07 | 69.67 ±0.08 | 73.47 ±0.12 | 71.00 ±0.23 | 83.53 ±0.25 |
| Mag    | 2.98 ±0.00 | **2124 ±63** | 75.48 ±0.14 | 73.43 ±0.35 | 75.49 ±0.06 | **77.19 ±0.16** | 85.74 ±0.20 |
| Sal    | 3.52 ±0.00 | - | 74.76 ±0.15 | 73.36 ±0.28 | 75.41 ±0.16 | 75.67 ±0.10 | 85.60 ±0.14 |
| Mag-G  | 5.68 ±0.10 | 2322 ±154 | **75.69 ±0.02** | 73.24 ±0.29 | 75.39 ±0.11 | **77.34 ±0.14** | 86.01 ±0.20 |
| STE    | **2.71 ±0.00** | - | 73.74 ±0.16 | **74.42 ±0.16** | **76.06 ±0.18** | 76.31 ±0.12 | **86.16 ±0.36** |
| Static | 3.21 ±0.02 | 1157 ±367 | 71.15 ±0.13 | 65.05 ±0.58 | 70.69 ±0.48 | 71.77 ±0.31 | 83.33 ±0.27 |
| SET    | 3.13 ±0.01 | **1723 ±414** | 74.12 ±0.07 | 69.83 ±0.56 | **75.47 ±0.54** | **76.40 ±0.18** | 84.20 ±0.12 |
| RigL   | **3.10 ±0.01** | 1535 ±434 | **74.51 ±0.11** | **71.10 ±0.32** | 75.52 ±0.27 | 75.46 ±0.05 | **84.64 ±0.14** |

Table 1: Performance of a selected subset of algorithms implemented in *JaxPruner* on a variety of benchmarks. We group algorithms that require storage or compute proportional to dense training in the middle and at the bottom group the fully sparse training algorithms. We report the validation accuracy for the image classification experiments (right). For T5-Base, we report per token cross entropy loss on the C4 validation split. DQN experiments report average returns on MsPacman environment. PViT corresponds to the ViT variant and training recipe suggested by [38].

|        | ViT-B16 | | ViT-B16+ | | PlainViT-S16 | |
|--------|------------|----------|------------|----------|------------|----------|
|        | Validation | Training | Validation | Training | Validation | Training |
| Dense  | 73.94 ±0.11 | **82.85 ±0.12** | 74.71 ±0.27 | **90.75 ±0.10** | **80.11 ±0.07** | **64.24 ±0.35** |
| Mag-G  | 73.24 ±0.29 | 75.57 ±0.26 | 75.39 ±0.11 | 80.96 ±0.13 | 77.34 ±0.14 | 57.37 ±0.62 |
| STE    | **74.42 ±0.16** | 78.01 ±0.22 | **76.06 ±0.18** | 85.46 ±0.05 | 76.31 ±0.12 | 54.86 ±0.41 |
| RigL   | 71.10 ±0.32 | 70.63 ±0.29 | 75.52 ±0.27 | 78.81 ±0.34 | 75.46 ±0.05 | 53.89 ±0.58 |

Table 2: Classification accuracies (%) of different recipes on ImageNet-2012 training and validation sets. The original ViT recipe for dense models leads to overfitting, while sparse networks achieve better generalization due to the regularization effect of sparsity.

uniform sparsity across layers and leave the first convolutional layer dense as recommended by Gale et al. [17]. Though most results match previous work, we observe a significant improvement for the accuracy achieved by the SET algorithm compared to the implementation done in [34].

We use a uniform sparsity distribution in our ViT experiments as we found using ERK distribution [18, 34] didn't lead to better results. Sparse vision transformers trained using the original recipe achieve better generalization even for the shorter, 90 epoch, training runs (ViT-B16). STE obtains the best results and exceeds the baseline performance by 0.2%. Interestingly, when we increase the number of training epochs to 300 (ViT-B16+), this gap widens and sparse ViT-B/16 trained with STE obtains 1.2% higher accuracy, despite having worse (higher) training loss. Dynamic sparse training methods (RigL and SET) perform poorly in shorter training runs, however with extended training, they achieve almost 5% higher accuracy and exceed the dense baseline.

Sparse models trained using the original ViT training recipe leads to better generalization despite having worse training performance (see Table 2). Next we train sparse models using the improved ViT recipe (PlainViT [38]), which achieves better generalization. Here validation performance follows training performance closely and the best pruning algorithm falls 3% short of the dense network.

## 4.2. Federated Learning and JaxPruner

Given the compute and communication constraints of the federated learning setting, pruning and sparse training are critical mechanisms to explore. FedJAX [6] supports federated learning research through JAX-based federated algorithm design and simulation, and can easily be integrated with *JaxPruner* to explore how to leverage sparse training in federated learning. In this paper we benchmark server-side pruning by using *JaxPruner* algorithms to change the server optimization step.

We test the effect of various pruning algorithms on the federated EMNIST character recognition benchmark [40], using the model architecture and task setup presented in Reddi et al. [41]. Pruning is applied on the server model on each round of training specified by the pruning schedule, before being broadcast to a sampled selection of clients to continue training. Each experiment is run for 1000 federated rounds, in which 50 clients are sampled and complete a single epoch of training on their data using a batch size of 32. For optimizers, we use SGD on clients and Adam on the server.

All pruning algorithms are configured with a target sparsity of 80%, the ERK distribution, an update frequency of every 10 rounds, and an update end step of 750 federated rounds. The sparse training algorithms (Static, RigL and SET) are configured to start updating in the first federated round, while the gradual pruning and straight through estimator algorithms are configured to begin pruning at round 250. All results reported are the average final accuracy on the evaluation dataset across five random trials. We find STE to perform best among the gradual pruning methods and RigL to outperform the other sparse training methods tested.

## 4.3. Language Modeling

We also build a *JaxPruner* integration with the *t5x* library [30], which opens access to a suite of Transformer-based [42] Language Models (LMs). In this section, we apply *JaxPruner* algorithms to a T5 encoder-decoder LM model [43].

Similar to experiments in Section 4.1, we prune 80% of the weights (5x compression) of our LM architecture. We train from scratch a T5-base (220M parameter) model to predict missing words within a corrupted span of text on the C4 dataset[3] with the Adam optimizer [48]. We report the per token cross-entropy loss on the validation split in Table 1. Our results show large differences in performance across the pruning algorithms. As in our ViT vision and federated learning experiments, STE outperforms other pruning algorithm and is within 5% of the dense baseline performance.

## 4.4. Deep Reinforcement Learning on Atari

Dopamine [31] library provides stable and comprehensive implementations for various Deep RL algorithms in JAX. We integrate *JaxPruner* with Dopamine as it has been used in the past for sparsity research [49, 50].

The Dopamine framework includes DQN [44], Rainbow [51], and other distributional deep RL agents like Quantile Regression for Distributional RL (QR-DQN) [52] and Implicit Quantile Networks (IQN) [53]. Though it is possible to run any of the Atari games [45] and agents, we choose MsPacman and DQN for our experiments.

We use the default hyper-parameter values provided in the Dopamine library together with the CNN architecture used in original DQN paper [44]. We apply sparsity to the existing model using the ERK distributions and at 98% target sparsity. We ran our experiments for 40M frames, 5 independent seeds and report the average returns calculated over 125000 environment steps at the end of the training.

## 4.5. Further Experiments

In this section we re-visit some of the common research questions in sparsity research and use *Jax-Pruner* to answer them. The experiments presented here only require a few lines of change in configurations.

**How do different sparsity structures affect pruning and sparse training?** Previous research has shown the importance of allowing full freedom to the algorithms when selecting parameters to prune. However such *unstructured sparsity* patterns are more difficult to accelerate compared to more structured sparsity patterns. *Block sparsity*, for example, removes weights in one or two dimensional blocks, leading to increased memory re-use and faster run-times [21]. Another more recent type of structured sparsity is called *N:M sparsity* [13], which allows at most N non-zero values over a one dimensional slice of M values. In *JaxPruner* such different sparsity structures are

---

[3]https://www.tensorflow.org/datasets/catalog/c4

| | Dense | Unstructured | 2:4 | 1:4 | 1:8 | 4x1 | 4x4 | 8x8 |
|---|---|---|---|---|---|---|---|---|
| Total Sparsity (%) | 0 | 80 | 49.9 | 74.9 | 87.4 | 79.8 | 79.7 | 79.9 |
| Gradual Magnitude | 73.94 | 73.43 | 73.29 | 72.35 | 44.96 | 69.28 | 67.61 | 68.57 |
| Magnitude STE | | 74.42 | 73.72 | 73.50 | 71.46 | 45.02 | 32.74 | 43.94 |

Table 3: ImageNet-2012 validation accuracies of sparse ViT-B/16 models with different sparsity structures after 90 epochs of training. N:M sparsity corresponds to the sparsity structure introduced in Mishra et al. [13], whereas NxM corresponds to block sparsity, which is applied along the first 2 dimensions. Total sparsities are slightly lower than the target since single dimensional variables are kept dense.

| | Dense | Embedding-Only | Encoder-Only | Decoder-Only | MLP-Only | All |
|---|---|---|---|---|---|---|
| Total Sparsity (%) | 0 | 8 | 27 | 44 | 36 | 80 |
| Validation Loss | 2.55 | 2.56 | 2.61 | 2.88 | 2.67 | 2.98 |

| Embedding Sparsity (%) | 80 | 90 | 95 | 98 | 99 |
|---|---|---|---|---|---|
| Total Sparsity (%) | 8 | 9 | 9.4 | 9.7 | 9.8 |
| Validation Loss | 2.56 | 2.59 | 2.58 | 2.61 | 2.66 |

Table 4: (top) Pruning different parts of the encoder-decoder T5-Base model to 80% sparsity. (bottom) Effect of increasing sparsity on embedding layers to the final validation loss. Both results are obtained using the STE method.

implemented through custom top-k functions [4] and can be configured easily. In Table 3, we ran experiments with these different sparsity structures (types) using 2 different algorithms: magnitude based STE and gradual magnitude pruning. Results shows that STE achieves good results for N:M, however performs poorly with block sparsity; for which gradual magnitude pruning achieves the best results.

**How does ERK compare to uniform sparsity distribution?** The sparsity level of each layer in *JaxPruner* can be configured through the sparsity distribution function. We provide implementations for 2 common distributions: (1) *Uniform*, which uses the same target sparsity for each layer (2) *Erdos-Renyi-Kernel* (*ERK*) [18], which adjust sparsity at every layer proportionally to the sum of its dimensions. We run these 2 distributions again by changing a single line in the configuration an share the results in Table 5. Unlike the results observed in previous work when using ResNet-50 models [34], Transformer models with ERK distribution don't achieve higher accuracy (ImageNet-2012) or significantly lower loss (C4) compared to a uniform distribution, highlighting an important area for future research.

| | Uniform | ERK |
|---|---|---|
| ViT-B/16 | 71.52 | 70.90 |
| T5-Base | 2.72 | 2.71 |

Table 5: Effect of uniform and non-uniform sparsity distributions in transformer models at 80% overall sparsity.

**Which layers of a transformer model are easier to prune?** Neural network architectures are often built from smaller building blocks and often multiple networks are combined for a specific purpose [54, 55]. When multiple types of layers, blocks and architectures are used together, looking solely at the shape of the parameters of each layer is not sufficient to determine the optimal sparsity. For example Graesser et al. [49] showed when training a reinforcement learning agent (SAC) [54], actor networks are significantly easier to prune than critic networks in the SAC agent. We do a similar study here with the T5-Base model trained on the C4 dataset. We prune 4 different parts of the T5-Base architecture in Table 4: (1) embedding layer. (2) Encoder blocks (self-attention and MLPs) (3) Decoder blocks (self/cross-attention and MLPs) (4) MLP layers after attention (both in encoder and decoder). Pruning different parts of an architecture, again, done through single line change in the configuration file by passing a `filter_fn` which decides which parameters to prune. We prune each layer to 80% sparsity, however since these different parts have different amounts of trainable parameters, models achieve different total sparsities. Though it is difficult to make a strict conclu-

---

[4]Sparsity pattern is calculated by selecting top-k scoring values using a given metric. We modify these top-k functions such that they follow the structure chosen.

sion due this, we observe embedding layer to be easy to prune. Therefore we perform additional experiments pushing the sparsity even further achieving 95% embedding sparsity with almost no performance drop.

# 5. Related Work

**JAX** JAX [3] is a Python library for high-performance machine learning research. With Autograd [56], JAX can automatically differentiate native Python and Numpy functions such as loops and branches in both forward and backward modes. Also, JAX supports just-in-time compilation of NumPy programs on multiple GPUs or TPUs in parallel with XLA [57]. Following the functional programming paradigm all transformations in JAX work on pure functions and thus can be composed together in an arbitrary order. Since these features can dramatically facilitate machine learning research, the community has started adopting JAX to develop new research frameworks in recent years, including for example Optax [10], FedJAX [6], Flax [5], JaxOpt [58], just to name a few.

**Hardware** There are numerous efforts towards hardware and software support for sparsity. An example of hardware acceleration for inference is the 2:4 fine-grained structured sparsity that was introduced by the Nvidia Ampere GPU series [28, 59]. When each contiguous block of four elements contains two zeros (viz. 50% sparsity), a low-overhead compression becomes possible which stores the non-zero values together with 2-bit indices. The hardware supports this compressed format by only operating on the nonzero values during the computation.

**Software** A promising direction for developing sparse software was pioneered for sparse linear algebra in the MT1 compiler [60] and generalized to sparse tensor algebra in the Tensor Algebra Compiler [61–63]. In these approaches, sparsity is treated as a property of tensors, not a tedious implementation detail, and a compiler automatically generates sparse code from a 'dense' definition of the computation where the programmer merely adds sparsity annotations to the tensor operands. A single description of a computation can be mapped to a wide range of sparse implementations, each tailored to specific sparsity properties. These ideas gave rise to, for example, sparse tensor support in the MLIR compiler-infrastructure [64] and proposed sparse extensions to JAX [3].

Sparse linear algebra binary libraries such as MKL [65] and cuSPARSE [66] implement sparse basic linear algebra subroutines for a small set of sparse data types. Generic libraries like Eigen [67] and CUSP [68] allow writing math-like expressions for a wider choice of data types. The Graph-BLAS [69] standard specifies a core set of general sparse matrix-based graph operations over arbitrary semi-rings. Many libraries implement this standard [70–73] for CPUs and GPUs. Libraries such as Sputnik [20], cuSPARSELt [74], and LIBXSMM [75] add new kernels and data types specific to deep learning, but still with limited portability. MegaBlocks [15] is a framework of efficient Mixture-of-Experts training on GPUs.

**Sparsity** Pruning and sparse training have witnessed a resurgence of research interests over the last few years, with many exciting developments and variations of standard sparsity approaches. See Hoefler et al. [47], Liu and Wang [76] for comparisons between various sparsity methods and a comprehensive analysis. Despite progress made, there is a need for sparsity libraries, benchmarks, and evaluation protocols. Gale et al. [17] compared few popular algorithms across different domains and architectures. Similarly [46] provided an extensive report on benchmarks used in pruning papers. One of the key existing libraries, OpenLTH [77], is primarily focused on easing research related to the Lottery Ticket Hypothesis [78], and facilitates implementation of magnitude-based pruning methods as well as computation of related metrics. Other mask-based pruning libraries in PyTorch include [79] and [80]. Alternatively, Ivanov et al. [81] focuses on providing acceleration in PyTorch through sparse matrix representations and operations. Many of these libraries have been used by many published papers. We hope our library would facilitate research in a similar way.

# 6. Conclusion

In this work we introduced *JaxPruner*, a new JAX library which aims to accelerate research in sparsity using 3 tenets: (a) fast integration (b) research first (c) minimal overhead. *JaxPruner* provides concise implementations of a diverse set of sparse training and pruning algorithms. Provided algorithms are easy to extend and support common sparsity distributions and structures. *JaxPruner*

also includes integration with other JAX libraries focusing four different domains. We benchmark our algorithms in these domains and share these baseline results alongside the library.

*JaxPruner* makes the implementation of new ideas and evaluation of them easy providing an excellent starting point for future sparsity research. Finally, though the primary goal of *JaxPruner* is to accelerate sparsity research, we believe its tenets and design can provide a blueprint for JAX libraries focusing different research domains.

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
