# OpenReview forum: "Jaxpruner: A Concise Library for Sparsity Research"
_CPAL.cc/2024/Conference — CPAL 2024 (Proceedings Track) Oral_

### Official Review · Reviewer_xdL7 · 2023-10-07

**Rating:** 7
**Confidence:** 5

**Review:**

The paper presents a JAX-based sparsity library for machine learning research. JaxPruner provide concise implementations of popular pruning and sparse training algorithms with minimal memory and latency overhead. The Optax gradient transformations for implementing algorithms is interesting. The paper is well written with appropriate code snippets to elucidate usage. Minimization of memory and run-time overhead by compressing binary masks for representing sparsity and incorporation of N:M sparsity are important features to have. However, I still feel the coverage of various popular pruning baseline algorithms are still missing (eg. SNIP, GrasP, SynFlow etc) if the focus is research. I find the further experiments section very interesting and informative. Some typos need to be fixed immediately (eg. Section 6- Concusion -> Conclusion). Overall, it is an important tool for sparse community and I recommend acceptance.

---

### Official Review · Reviewer_m57v · 2023-10-07
**Good Library of Jax for Network Pruning**

**Rating:** 6
**Confidence:** 5

**Review:**

This paper presents a new neural network pruning library implemented with Jax, called JaxPruner, with 3 tenets: (a) fast integration (b) research first (c) minimal overhead. The library has implemented the major components of network pruning, supporting different pruning criteria, schedules, etc. They also establish the results of many baseline pruning schemes. This work does not introduce new pruning algorithms. Instead, they offer a new pruning library with Jax, which could benefit the Jax & pruning community.

Pros:
1. First and foremost, there is no prevailing Jax library for neural network pruning. This paper bridges this gap, which could benefit many researchers, esp. those using Jax as their deep learning framework and are interested in pruning.
2. JaxPruner is featured by fast integration, research first, and minimal overhead, which could make the library easy to use.
3. JaxPruner also provides strong baselines in either traditional pruning (pruning a pretrained model) or sparse training (pruning at initialization).

Cons:
1. Some of the results look interesting but lack enough discussion. E.g., in Tab. 1, for the row ViT-B16+, SET and RigL beat Dense with the prolonged training, while underperforming Dense in the row ViT-B16. This is quite unusual. Did the authors double-check and confirm the results are correct? If so, why? More explanations or discussions are highly suggested. Unreliable baseline results could mislead the community.

2. The library implemented many baseline pruning schemes like RAND, SAL, and MAG -- these are different pruning criteria. There is another major group of pruning methods that use sparsity-inducing penalty terms for sparsity, such as [*1-*4]. I was wondering whether this group of methods can easily fit into JaxPruner?

3. Typos: Line 258. with block sparsit; -> sparsity.

- [*1] 2016-NeurIPS-Learning Structured Sparsity in Deep Neural Networks
- [*2] 2018-ICLR-Learning Sparse Neural Networks through L0 Regularization
- [*3] 2021-ICLR-Neural Pruning via Growing Regularization
- [*4] 2022-ICLR-Dual Lottery Ticket Hypothesis

---

### Official Review · Reviewer_Kqti · 2023-10-07

**Rating:** 8
**Confidence:** 4

**Review:**

## Overview

The authors introduce a new sparsity-focused library tailored for machine learning research. Built on the JAX platform, the library facilitates the implementation of an array of sparse algorithms, encompassing various sparsity patterns, both static and dynamic training sparsity, and multiple pruning metrics. A comprehensive set of experiments spanning diverse domains such as image recognition, natural language processing, federated learning, and deep reinforcement learning, attests to the library's adaptability.

## Comments

(+) The library has been effectively tested on an array of tasks, from image recognition to deep reinforcement learning, and incorporates numerous sparsity methodologies, from unstructured to dynamic sparsity. This demonstrates the library's comprehensive adaptability.

(+) Some features, like using int8 type for storing masks, are commendable, particularly given the growing prominence of expansive foundation models that demand extensive mask memory.

(+) The paper is well organized, delving initially into the reasons for creating a new sparsity-centered library, transitioning into its primary features, and culminating with in-depth benchmarking results.

(+) This library makes a notable contribution to the sparsity community, simplifying the process of implementing new ideas.

(-) An elaboration on its compatibility with other sparsity training models, like Mixture-of-Experts, would be beneficial.

(-) The paper lacks a comparative analysis with prevalent libraries (w. Pytorch or Tensorflow), particularly concerning training/inference velocity and memory consumption.

---

### Meta-Review · Area_Chair_EENM · 2023-11-12

**Recommendation:** Accept (Oral)
**Confidence:** 4

**Metareview:**

This submission presents a sparsity-centric JAX library designed for machine learning research. Leveraging the JAX platform, this library streamlines the development of a wide range of sparse algorithms, accommodating diverse sparsity patterns, including both static and dynamically evolving training sparsity, as well as incorporating multiple pruning metrics. All authors voted for the acceptance. I also see it will benefit the sparsity community a lot.

---

### Decision · Program_Chairs · 2023-11-19

**Decision:**

Accept (Oral)

**Comment:**

The paper introduces JaxPruner, a JAX-based sparsity library for machine learning research, focusing on neural network pruning. It provides concise implementations of various pruning and sparse training algorithms, with minimal memory and latency overhead. Reviewers commend the library's comprehensive adaptability and features, such as using int8 type for storing masks, which is particularly useful for large foundation models. However, they suggest providing more information on compatibility with other sparsity training models and conducting a comparative analysis with prevalent libraries like PyTorch or TensorFlow, especially concerning training/inference velocity and memory consumption. Additionally, one reviewer raises questions about the reliability of certain baseline results and suggests more detailed explanations or discussions for unusual findings.

The action PC chair for this paper is Atlas Wang, who made the decision after carefully reading the paper as well as the comments by all reviewers and AC. The decision is agreed by all PC chairs.